# Exploring Associated Factors of Subjective Health Literacy in School-Aged Children

**DOI:** 10.3390/ijerph17051720

**Published:** 2020-03-06

**Authors:** Alexandra Fretian, Torsten Michael Bollweg, Orkan Okan, Paulo Pinheiro, Ullrich Bauer

**Affiliations:** 1School of Public Health, Bielefeld University, 33615 Bielefeld, Germany; 2Centre for Prevention and Intervention in Childhood and Adolescence (CPI), Faculty of Educational Science, Bielefeld University, 33615 Bielefeld, Germany; torsten.bollweg@uni-bielefeld.de (T.M.B.); orkan.okan@uni-bielefeld.de (O.O.); paulo.pinheiro@uni-bielefeld.de (P.P.); ullrich.bauer@uni-bielefeld.de (U.B.)

**Keywords:** subjective health literacy, children, social gradient, motivation, HLS-EU-Q

## Abstract

Low health literacy is considered to lead to worse health-related outcomes and behaviors and has therefore been recognized as a social determinant of health. While health literacy and its potential determinants have been studied in adults, little research has been conducted with children. This study aims to address this research gap by investigating factors associated with children’s subjective health literacy. Cross-sectional data was collected from fourth graders at German schools with a self-report questionnaire. Sociodemographic characteristics, health-related attitudes, and motivation were analyzed. We used hierarchical multivariate linear regression to explain variance in the dependent variable “subjective health literacy”. A total of n = 907 fourth graders were surveyed. Regarding health literacy, eight out of ten participants (82.2%) reported that it was “rather easy” or “very easy” to deal with health-related information. Family affluence, but not language spoken at home, was significantly related to subjective health literacy, after controlling for confounding. Moreover, parental health orientation, self-efficacy, and motivation are factors significantly associated with health literacy. Based on the results of this study, it is hypothesized that a general motivation to learn new things about health, as well as an environment promoting health-positive behavior, might foster children’s health literacy.

## 1. Introduction

Health literacy has been acknowledged as a determinant of health [1,2] and a critical means of increasing health equity [3]. It is considered to be a modifiable factor that can be addressed through education [2,3,4] and may play a key role in interventions aiming to prevent disease and promote good health. Thus, the potential of investigating and fostering health literacy at an early age has been emphasized [5,6,7]. Nevertheless, up to now little is known about the health literacy of children and adolescents [8], how it can be promoted, and which groups of young people are likely to have low health literacy.

The conceptual heterogeneity in health literacy research plays a role in the field’s relatively slow progression in terms of generating comparable data [9]. Some core components, however, are consistently mentioned within the health literacy discourse. One attempt to unify the definitions and point out the commonalities between them was made by Sørensen [10] (p. 3), who put forth the following overarching definition:

“Health literacy is linked to literacy and entails people’s knowledge, motivation and competences to access, understand, appraise, and apply health information in order to make judgments and take decisions in everyday life concerning healthcare, disease prevention and health promotion to maintain or improve quality of life during the life course.” Based on this definition, our study focuses on subjective health literacy as an inherently dynamic and multidimensional construct. This understanding is widely used in public health and has been developed independent of and in opposition to the initial medical understanding of health literacy, comprising mainly functional skills [11]. Various studies with adults, coming from both the medical and the public health fields, have demonstrated that health literacy follows a social gradient, expressed by lower levels of health literacy among minorities [12,13], groups with low socio-economic status [13,14,15,16,17,18], and less educated populations [13,14,15,17,18,19]. A similar trend can be observed in younger age groups. Adolescents’ health literacy, for instance, has been found to be associated with indicators of social position, such as socioeconomic status and parental education [20,21,22,23,24]. When it comes to young people aged 15 to 25, a study showed that migration background, gender, and the educational level of the participants and their parents explained 7% of the variance in health literacy [23]. Similarly, family wealth, together with the regions in which the 15-year-old adolescents reside, and parents’ educational levels were able to explain 3.1% of the variance in health literacy [24]. Moreover, a systematic review identified a link between low parental health literacy and worse health outcomes in children [25]. Empirical evidence regarding the determinants of health literacy among younger populations is sparse, as only few studies have investigated and reported on potential social determinants of children’s health literacy. When focusing on younger age groups, especially children aged 12 years and younger [26], a handful of studies can be named that describe different associated factors of health literacy or closely related measures (e.g., health knowledge, health-related attitudes). 

Differences across gender and age have been observed with regard to health-related attitudes [27], as well as better health knowledge and more prevalent healthy practices among elementary school students (compared to middle school students) and students living in more affluent regions [28]. In a study conducted by Yu [28], girls were shown to have better health knowledge. Driessnack [29] found higher levels of functional health literacy among older children, but no differences related to family socioeconomic level or ethnicity. Lastly, Schmidt and colleagues [30] report that no effect of household income on children’s communication and knowledge was observed, but that children of parents with a higher educational background, as well as girls, had more knowledge and talked more about health-related topics. While these studies offer interesting insights, none tackle the comprehensive, multidimensional understanding of subjective health literacy that goes beyond basic literacy and numeracy skills, including varying capabilities that support health-related decision making [31]. To our knowledge, the only other study measuring a comprehensive understanding of health literacy in nine- to ten-year-old children was conducted by Teufl [8], although they have not published any information on associated factors of health literacy yet. 

The aim of this study is to explore potential determinants of subjective health literacy in children and verify whether there are indications of a social gradient in fourth graders (based on family affluence) as this has been observed in many adult samples. Health literacy has been operationalized on the basis of the comprehensive model developed by Sørensen [10], but adapted to nine- and ten-year-old primary schoolchildren in Germany. In our analyses, we focus on the relative importance of the factors potentially associated with health literacy in this target group, such as family affluence, migration background, gender, attitudes, and functional health literacy. 

## 2. Materials and Methods 

### 2.1. Study Design

The current investigation is based on a cross-sectional study in which health-related data were obtained from fourth grade schoolchildren as part of a study validating a newly developed measure of children’s subjective health literacy. The study was conducted within the scope of the research project “Methods of Measuring Health Literacy of Children” (MoMChild), which is described elsewhere in more detail [32].

### 2.2. Data Collection, Recruitment, and Sample

Data collection, data entry, and recruitment of the participants was carried out by the “Sozialwissenschaftliches Umfragezentrum Duisburg”, a sub-contracted institute specialized in conducting surveys. All analyses were conducted by the authors. The fourth grade classes of primary schools were invited to participate in the study, with at least one fourth grade class from each school participating. 

A quota sampling procedure was used to obtain a similar distribution of schools located in rural versus urban areas, with larger versus smaller populations with backgrounds of family migration. In other words, the same share of schools located in rural areas with high and low migration background, respectively, as well as in urban areas with high and low migration background, respectively, were invited to participate. The data was collected between November 2016 and June 2017 in primary schools located in the federal state of North Rhine-Westphalia (NRW), Germany.

A total of 32 out of 200 invited schools participated, each with at least one fourth grade class or at most four fourth grade classes. From 67 surveyed fourth grade classes, a total of 907 children participated in the survey. The response rate within classes ranged from 26% to 92%, with an average of 61%. For the analysis, eight cases were removed due to potential bias (e.g., teachers helping them complete the questionnaire) and incomplete data (e.g., not finishing the questionnaire due to health issues). The self-reported questionnaire was administered during two lesson hours in a paper-and-pencil classroom survey and took on average 34 (SD = 9) minutes to be completed. Standardized instructions for completing the questionnaire were provided to all participants by trained staff. Data was collected from children only.

Parental informed consent was requested prior to study enrollment. Children’s participation was voluntary and participants were free to quit at any time. No personal data, such as name or place of residence, was collected. The conduct of the survey was approved by the Ethics Committee of Bielefeld University (No. 2016-141-R). Data protection procedures were developed in cooperation with Bielefeld University’s data protection office and are in line with the guidelines of the German Psychological Society. No incentives were used in this study.

### 2.3. Measures

#### 2.3.1. Subjective Health Literacy

Health Literacy is the main dependent variable in the analyses presented here. It was measured with an age-adapted version of the European Health Literacy Survey Questionnaire (HLS-EU-Q) [33], the HLS-Child-Q15-DE. The questionnaire was adapted specifically for fourth graders. The development and validation of the instrument are reported elsewhere [34,35]. The HLS-Child-Q15-DE consists of 15 items assessing the perceived ease or difficulty in finding, understanding, appraising, and applying health-related information. All items are phrased “how easy or difficult is it for you to…”. Responses are given on a four-point Likert scale ranging from 1 = “very difficult” to 4 = “very easy”. In line with the original HLS-EU-Q study [36], a mean score is calculated for participants with a maximum of 20% missing responses (i.e., a valid response for at least 12 out of 15 items). Higher scores correspond to perceived ease in dealing with health information (i.e., higher “subjective health literacy”).

A number of predictor variables were included in this study:

#### 2.3.2. Demographic Variables

Children’s age, gender, the language spoken with parents, and family affluence were recorded. Age was dichotomized as “9 years or younger” vs. “10 years or older”, as most fourth grade children in Germany are nine or ten years old, with few exceptions. Gender was queried with the item “Are you a girl or a boy?” As an indicator for a family background of migration, the language spoken at home was collected for both parents or caretakers and dichotomized as “exclusively German” vs. “not exclusively German”. As data were collected from children only, it was not deemed reliable to record household income as well as parents’ occupation and education level, which are standard indicators for socio-economic status. Instead, family affluence, a proxy for socio-economic status, was captured with the Family Affluence Scale (FAS) III [37], an indicator of material affluence widely used in health inequality research. It includes six items on material assets such as the number of cars or bathrooms in the household. For those children validly answering at least four of the questions, the sum scores were computed ranging from 0 (lowest affluence) to 13 (highest affluence).

#### 2.3.3. Functional Health Literacy

Children’s *functional health literacy* was assessed with a cloze procedure reading comprehension test, which was created for the target group. The format of the comprehension test was inspired by the “Test of Functional Health Literacy in Adults” [38] which assesses patients’ literacy skills regarding written information materials that they would realistically have to deal with in a healthcare setting. More specifically, the instrument used here is based on a text regarding vaccination that was previously developed for a health-related website for children. In the text, 12 words are missing, and children are asked to choose the alternative which logically fits in the blank. For instance, one of the sentences reads as follows: “You can ____ a vaccine against many ____”, whereby “do”, “must”, “can”, “**get**” are offered as potential answer options for the first blank, and “accidents”, “**diseases**”, “waiting rooms”, and “cats” for the second (translated from German for illustrative purposes). The correct answers have been printed here in bold. One point is awarded for each correctly chosen word, with a maximum sum score of 12. Most blanks can be filled in relatively easily with average language skills.

#### 2.3.4. Perceived Parental Health Orientation

“*Perceived parental health orientation*” was captured through a combination of two questions: “My parents make sure that I eat healthy”, and “It’s important to my parents that I exercise regularly”. Responses were provided on a four-point Likert scale ranging from 1 = “fully disagree” to 4 = “fully agree”. As both items showed ceiling effects (95.5% and 83.5% agreement, respectively), items were dichotomized as “full agreement” vs. “partial agreement or disagreement”. Through this procedure, more variance could be retained, and both dichotomized items showed a higher correlation with the original items (r = 0.90 and 0.87, respectively; *p* < 0.001), compared to dichotomizing “agreement” vs. “disagreement” (r = 0.64 and 0.78, respectively; *p* < 0.001). The composite indicator “perceived parental health orientation” was dichotomized as “very high” for participants stating full agreement to both individual items vs. “not very high” for all others.

#### 2.3.5. Self-Efficacy

Children’s *self-efficacy* was assessed through a single item: “For most problems, I can find a solution”. Recorded on a four-point Likert scale, the answers comprised 1 = “*not true at all”,* 2 = “*rather untrue*”, 3 = “*rather true*”, and 4 = “*completely true*”. Again, ceiling effects were observed (85.8% “rather true” and “true”) and the item was dichotomized as “full agreement” vs. “partial agreement or disagreement”.

#### 2.3.6. Motivation

To also include motivational aspects into the statistical analysis, the following statement was used: “I like learning something new about health”. Again, the answers were recorded on a four-point Likert scale, which comprised 1 = “*not true at all*”, 2 = “*rather untrue*”, 3 = “*rather true*”, and 4 = “*completely true*” as possible response options. Due to ceiling effects (80% “rather true” and “true”) this variable was also dichotomized as “full agreement” vs. “partial agreement or disagreement” which was, again, supported by a high correlation with the original item. 

To describe the reliability of the scales, Cronbach’s α and Kuder–Richardson-20 (KR-20) have been used.

### 2.4. Statistical Analyses

As data was collected from school classes, the potential clustering of health literacy scores at the school and classroom level was examined by calculating the intraclass correlation coefficient. The result turned out to be non-significant, indicating the absence of clustered data. Thus, no multi-level analyses were necessary. 

Frequencies and percentages were used to describe sample characteristics. The association between health literacy and potentially related factors was explored through bivariate analysis, using the t-test for the association with relevant dichotomous variables and the Pearson correlation coefficient for the association with functional health literacy and FAS. 

A hierarchical linear regression was chosen to model the relationships between health literacy scores (dependent variable), in which the following models were tested:Model 1: demographic characteristics (gender, age, home language, FAS);Model 2: individual characteristics (functional health literacy, self-efficacy, interest in learning something new about health);Model 3: contextual characteristics (parental health orientation).

All of the previous independent variables were included in the subsequent models. Hierarchical regression was chosen in order to reveal the relative importance of the different associated factors of health literacy in childhood. The rationale behind choosing three models was the possibility of observing differences in explained variance and to test the efficacy of more complex models in comparison to simpler ones. The first model only included demographic and socio-economic indicators; the second model incorporated variables which have typically been described as constituents of or closely related to health literacy. Both functional health literacy and motivation (here: interest in learning something about health) are described by Sørensen [10] as being either part of or closely associated with health literacy. In contrast, self-efficacy was included as a potential confounder based on the form of enquiry that is used to assess health literacy itself (“how easy or difficult is it for you to…”). In addition, the third model indirectly captures parental attitudes towards health, which may constitute a major promoting or hindering factor for the development of health and health literacy in the home environment. This acknowledgement of contextual factors seems relevant in the face of the debate on children living in dependence of their parents vs. children being autonomous actors [31].

Key assumptions for performing the multiple linear regression analyses were tested before computing the models; homoscedasticity was verified through visual examination of the graph of standardized residuals against standardized predicted values. The absence of multicollinearity was tested through the variance inflation factor (VIF) < 5.00, the absence of autocorrelation was verified through the Durbin–Watson test, with values from 1.5 to 2.5, and the absence of influential observations was tested through Cook’s Distance < 1.00.

SPSS (IBM Corp, Armonk, New York, USA) Statistics [39], version 25, was used for the analyses. The criterion for statistical significance in all analyses was a p-value of maximally 0.05. 

## 3. Results

### 3.1. Sample Characteristics

The sample displayed a balanced distribution in terms of age and gender. About one third of the participants spoke a language other than German at home, indicating the possible presence of a family background of migration. Further sample characteristics can be found in Table 1.

For the FAS, 99.1% valid composite scores were obtained, ranging from one to 13 (mean = 8.61, SD = 2.34). Most children had a family wealth score of nine while more than half of the sample (54.4%) had a score of nine or higher. 

Functional health literacy was high in this sample. On a range from zero to 12 points, children obtained a mean score of 9.6 points (SD = 2.2). Most of them achieved eleven points. The internal consistency of the functional health literacy items was acceptable (KR-20 = 0.72). 

Health literacy showed small, significant correlations with functional health literacy (r = 0.10, *p* < 0.01) and FAS (r = 0.15, *p* < 0.01).

### 3.2. Health Literacy and Its Associated Factors

More than nine in ten children (93.5%) provided valid answers for at least 13 out of 15 items in the HLS-Child-Q15-DE, making it possible to calculate the health literacy mean score. On average, these children perceived dealing with health-related information as “rather easy” (mean = 3.34, SD = 0.37). Moreover, eight out of ten participants (82.2%) reported that it was “rather easy” or “very easy” to deal with health-related information. The health literacy scale demonstrated good internal consistency (Cronbach’s α = 0.79). Table 1 shows the mean differences in health literacy across sample characteristics. 

All assumptions for conducting multiple linear regressions were met. The three models of the hierarchical linear regression are presented in Table 2. The first regression model includes relevant background characteristics of the participating children, namely gender, age, language spoken with parents and family affluence. In this model, which explains 2% of variance in health literacy scores, family affluence was the only significant predictor. The second and third model were able to provide greater explanatory value. Functional health literacy together with self-efficacy and motivation explained an additional 14.4% in variance in health literacy scores, while in the third model, perceived parental health orientation explained an additional 2.9% in variance. 

Altogether, age, gender, language spoken with parents, family affluence, functional health literacy, parental health orientation, self-efficacy and motivation explained 19.3% of the variance in health literacy. In the final model, family affluence, functional health literacy, self-efficacy, motivation, and parental health orientation are significant predictors of subjective HL, while all other variables are not. 

When considering the three most influential individual predictors, one can note that the highest (standardized) regression coefficients have been registered for motivation, parental health orientation, and self-efficacy. For example, children that report “very high” parental health orientation score 0.13 units higher on the HL scale than children who do not (if all other variables remain constant).

## 4. Discussion

At the present, there is a paucity of research examining the health literacy of young children. Accordingly, the study described here constitutes an important contribution to the evidence base on the health literacy of this under-researched target group. We were able to explore the relative importance of several factors associated with health literacy among fourth grade schoolchildren. 

### 4.1. The Health Literacy of Children

First, we would like to align our results with other relevant findings related to children’s health literacy or health-related attitudes and knowledge. The diversity in conceptualization and operationalization, however, makes comparison difficult and, thus, rather speculative. 

The results presented reveal that, on average, the subjective health literacy of children in our sample is rather high, with eight out of ten participants (82.2%) reporting dealing with health-related information as “rather easy” or “very easy”. Similarly, Brown [27] assessed whether nine- to 13-year-olds consider understanding most health-related information easy or difficult. The self-report indicated that 78.1% of the children indicated it to be “very easy” or “sort of easy,” which is in line with our findings.

Compared to a study aimed at investigating the comprehensive health literacy of nine to 13-year-old children [30], the findings seem conflicting at first glance. The study investigated knowledge, communication, attitudes, behavior, and self-efficacy as separate dimensions of health literacy. The scores on health attitudes were found to be rather high (mean = 14.5; SD = 2.48; range 3–16), similar to our findings. However, the results diverged when considering the health knowledge dimension, which appeared to be rather limited (50.8% had one out of four correct, 38.0% had two out of four) [30]. This difference might be based on the measurements used but could also stem from differences between samples. Thus, even though both studies had a similar understanding of health literacy, the results are difficult to compare, given the differences in operationalization. The lack of unanimity in defining and measuring health literacy continues to impede comparability [9].

When trying to put the results into a life course perspective, an attempt to draw comparisons to the subjective health literacy of older age groups should be made. It has previously been observed that subjective health literacy declines with old age [16]. This can be interpreted as an indicator of an increasingly better—i.e., a more realistic—judgement of the challenges that individuals face when dealing with their own health and the healthcare system. Thus, the comparatively lower burden of disease in younger people and their accordingly more limited experience navigating the health care system or handling disease may prompt them to think that it is easy to do so. 

### 4.2. The Associated Factors of Health Literacy in Children

Second, we would like to discuss and interpret the associated factors identified in relation to subjective health literacy. In the first regression model, the sociodemographic indicators age, gender, home language, and family affluence were tested. Although differences in health literacy with regard to gender, age, and family background of migration have been observed [16], this was not the case in our study. In the case of the participants’ age, this is not surprising, as the age range was small (nine-year-olds and younger vs. ten-year-olds and older). Thus, further studies with broader age ranges are needed to investigate whether and to what extent there are differences in subjective health literacy in different phases of childhood and adolescence, as has been indicated by other studies [27,28,29]. 

Regarding gender, it might be plausible that in our sample there had not yet been any differential development of health literacy among the boys and girls surveyed at that point in time. Some studies [27,28,30] have, however, observed a significant difference in the health literacy of boys and girls in similar age groups. Accordingly, it is worthwhile to investigate at which age, under which circumstances, and regarding which components of health literacy (e.g., subjective, knowledge, etc.) gender-related differences emerge.

The indicator of a family background of migration used in this study (home language) identified 33.6% of the sample as having a family background of migration, compared to 43.6% of all students attending primary school in the federal state of North Rhine-Westphalia (94.2% of which with a parent not having been born in Germany) [40]. Accordingly, it is possible that the indicator home language underestimates the number of children with a migration background, which could be reasonably explained by their parents speaking exclusively German at home (false negative). No significant differences in health literacy related to home language were found in our study. This indicates that migration background, as assessed by this proxy measure, may not—at least not yet in this age group—affect the development of health literacy. 

In this study, we found that children with higher family affluence perceived dealing with health information to be significantly easier than their less well-off peers did. Thus, there is evidence suggesting that a social gradient of (subjective) health literacy is present in children as young as nine to ten years and that these differences may even emerge earlier in the course of their life. This social gradient has also been observed by Yu [28] among children as well as in adolescents. In contrast, Schmidt and colleagues [30], in their study with children, have not found evidence of a social gradient of health literacy related to income. Although there seems to be evidence indicating the presence of a social gradient of health literacy among young people, it may be worthwhile to further investigate this subject matter in different samples, also considering potential differences between indicators such as family affluence and income. Overall, gender, age, home language, and family affluence were able to explain no more than about 2% of variance in health literacy scores.

The second regression model, in contrast, revealed that significantly more variance in subjective health literacy scores (14.4%) could be explained by the combination of functional health literacy, self-efficacy, and motivation. While these factors have been discussed as being related to or part of health literacy [10], to our knowledge, this is the first study to concurrently investigate these factors and quantify their relative importance in explaining variance in children’s subjective health literacy. It is plausible that functional health literacy is associated with subjective health literacy, as a low level of functional health literacy may be a significant barrier when children try to access, understand, appraise, and apply health-related information. 

Self-efficacy had been hypothesized to be related to the measure of health literacy that we employed, which is supported by our data. This indicates that self-efficacy, on the one hand, might be a source of bias—i.e., part of what is measured by subjective health literacy may be equivalent to self-efficacy—and/or, on the other hand, might be a powerful facilitator in dealing with health information. Empirical evidence in adults supports the latter assumption [41]. Based on these considerations, the role of self-efficacy needs to be further investigated. If being ruled out as a confounder, self-efficacy could pose a promising and feasible approach to empowering people to successfully deal with health information.

Motivation regarding the acquisition of health information, namely the item “I like learning something new about health” proved to be the most potent independent variable in our analysis. This finding can certainly inform further studies and interventions, as it shifts the focus to motivational aspects of dealing with health information instead of a person’s cognitive capability. Thus, specifically in the educational setting, the transmission of knowledge might not be enough when questions like “do students *want* to learn?” and “*what* do they want to learn about?” are neglected. As our data indicate, motivation could contribute to the perceived manageability of health-related tasks. Nevertheless, promoting reading and writing capabilities, as well as the transmission of factual knowledge around the topic of health (i.e., functional health literacy), still remains a core responsibility within the school context and might have an independent impact when it comes to promoting the health literacy of children. 

In the last regression model, perceived parental health orientation is included as a contextual factor of health literacy. Through this, the explained variance in health literacy scores can be increased to 19.3%. This highlights the need for future studies to consider novel indicators of participants’ proximal social surroundings, because, as our data shows, one contextual factor as simple as perceived parental health orientation was able to explain as much variance in the outcome variable as the standard indicators age, gender, home language, and family affluence combined. While this observation may differ in other samples (e.g., in a sample with a wider age group), it will still be interesting to investigate the effect that different contextual indicators—parental education, income, or occupation, but also parental attitudes, habits, or health literacy—have on the development of children’s health literacy in different stages of youth.

### 4.3. The Social Gradient

Third, we would like to focus on structural inequalities when interpreting our findings.

Family affluence was the only significant variable in the first model and remained significant even when controlling for functional health literacy, self-efficacy, and health-related motivation in the last model. While material living conditions and socio-economic status have manifold implications for healthy child development [42,43] and seem to play a role in shaping health literacy of children at the same time [44], it is crucial to tackle the problem at its roots. Structural inequities affecting parenting and living conditions should be addressed at the policy level in order to mitigate such effects. However, this recommendation is beyond the scope of this publication and likely applies to the greater number of findings in health inequity research. 

In this sample, the social gradient in health literacy is not only conspicuous with regard to material living situations, but also in more discrete, complex structures of social reality—e.g., parental health orientation. While family affluence was able to explain 2% of variance in health literacy, parental health orientation explained 2.9% of variance. Parental health orientation was also the second strongest associated factor of health literacy, indicating that family support and encouragement of healthy behaviors, such as practicing sports and eating healthily, may foster health literacy. However, parental attitudes, cultural values, their orientations and further resources vary along a social gradient and are, of course, affected by their socioeconomic status, making it likely that parental health orientation is affected by the family’s social position [45]. 

While parents may be important actors fostering a positive orientation towards health in children’s lives, it can be argued that other actors and settings that children are part of play a similar role. For example, results from a Finnish study show that higher health literacy is significantly associated to membership in a sports club [46]. Alongside practicing physical activity, sports clubs may promote positive attitudes and self-determined motivation for physical activity [47]. Future studies should investigate whether and how positive attitudes towards health (e.g., healthy eating or practicing physical activity) promoted within a particular setting (e.g., family, school, or sport clubs) facilitate or mediate the development of children’s health literacy. 

### 4.4. Limitations

Given that, on average, the sample in question had high subjective health literacy, displayed a rather high family affluence, good functional health literacy, and generally tended to select positive answer options, it is possible that selection bias may have affected the survey. Selection bias may have had an influence at various levels—at the school, class, and individual levels. Given the response rate of 16%, it could be that more schools interested in the topic of health participated. At the same time, due to social desirability, it is possible that school principals assigned the survey to classes known for having students with higher educational attainment. At the class level, another selection process may have taken place, insofar as only those children that were willing and had a letter of consent from their parents participated in the survey. Since the letter of consent was only provided in German, it represents a potential barrier for parents who do not speak German (proficiently). This may have especially hindered the participation of children who only speak a language other than German. 

Moreover, given that the sample is not a representative one, it remains unclear whether these findings can be generalized to other samples. Although the quota sampling procedure aimed for a certain amount of variability in terms of school location (areas with high vs. low migration background as well as rural and urban areas), many responses were rather skewed in a positive direction. It is unclear whether this issue arose in this particular sample, or whether the discriminatory power of the items was insufficient. Thus, replicating the study findings in a representative sample of schoolchildren is necessary. 

Further, all data is solely based on children’s self-reports. While it is crucial to include the perspectives and opinions of children in research, a comparison with “objective”, i.e., performance-based, data would potentially provide a more comprehensive understanding on the issue at hand. It would be relevant to determine how perceived vs. objective measures influence each other and what relative importance they have in explaining differences in health literacy. Therefore, an objective measurement of parental socio-economic status and migration background, as well as the self-reported health orientation of parents could be included to clarify this. 

In terms of measurement, another potential shortfall must be mentioned. For the variables self-efficacy and motivation, we used single-item indicators, and for parental health orientation, we used two combined items instead of a validated scale. There may be a risk that single items do not assess the underlying construct as well as scales. However, at least in the case of self-efficacy, a single-item measure is not unheard of and has proven to be a reliable alternative to a 20-item scale, even displaying a better predictive value at follow-up than the scale as a whole [48]. This indicates that it might be feasible using single-item instruments. However, in general, it seems desirable to use validated scales, at least until more economic measures have proven to be as reliable.

Finally, given the cross-sectional nature of this study, no inferences regarding causality can be drawn. Therefore, the recommendations for practice, e.g., fostering interest and motivation towards health-related behaviors at home or at school in order to increase health literacy, need to be regarded as hints and suggestions which will have to be verified or falsified through further research. Longitudinal and intervention research is needed to verify the assumptions drawn.

## 5. Conclusions

This is one of the first studies to take a multidimensional approach in exploring potential determinants of health literacy in youth. Not only were already well-established associated factors of health literacy investigated, namely, age, gender, family affluence, migration background, and functional health literacy, but also motivational and contextual aspects, such as self-efficacy, motivation, and parental health orientation. Based on the results of this study, it can be hypothesized that the motivation to learn something new about health, as well as a supportive environment in which health-promoting behaviors are endorsed, could benefit the health literacy of children. Future interventional research should verify whether cultivating health in the home environment as a valuable asset to be gained and maintained, can indeed contribute to the development of health literacy. 

## Figures and Tables

**Table 1 ijerph-17-01720-t001:** Sample characteristics and health literacy mean scores based on the data of the MoMChild study [32].

Indicator	n (%)	mean HL (SD)
total	899 (100)	3.34 (0.37)
Gender		
Female	480 (53.4)	3.33 (0.39)
Male	404 (44.9)	3.35 (0.35)
Missing	15 (1.7)	
Age		
8−9	434 (48.3)	3.35 (0.33)
10−12	457 (50.8)	3.33 (0.40)
Missing	8 (0.9)	
Language spoken with parents		
Exclusively German	578 (64.3)	3.33 (0.36)
Not exclusively German	300 (33.6)	3.37 (0.38)
Missing	19 (2.1)	
Perceived parental health orientation		
Very high	390 (43.4)	3.25 (0.36)
Not very high	437 (48.6)	3.45 (0.35) ***
Missing	72 (8.0)	
“I can find a solution for most problems”		
Full agreement	325 (36.2)	3.47 (0.34)
Partial agreement or disagreement	519 (57.7)	3.27 (0.37) ***
Missing	55 (6.1)	
“I like learning something new about health”		
Full agreement	362 (40.3)	3.48 (0.33)
Partial agreement or disagreement	475 (52.8)	3.24 (0.37) ***
Missing	62 (6.9)	

Notes: *** indicates significant differences in health literacy scores between the respective groups on the level of p < 0.001.

**Table 2 ijerph-17-01720-t002:** Regression models based on the data of the MoMChild project [32].

	Model 1			Model 2			Model 3		
Variables	B (SE B)	*β*	*p*	B (SE B)	*β*	*p*	B (SE B)	*β*	*p*
Age	−0.01 (0.03)	−0.01	0.924	−0.01 (0.03)	−0.01	0.858	−0.01 (0.03)	−0.01	0.871
Gender	0.01 (0.03)	0.02	0.674	0.04 (0.03)	0.05	0.125	0.04 (0.03)	0.05	0.115
Language spoken with parents	0.05 (0.03)	0.07	0.075	0.05 (0.03)	0.06	0.077	0.04 (0.03)	0.06	0.103
Family affluence	0.02 (0.01)	0.15	0.000 ***	0.02 (0.01)	0.12	0.001 ***	0.02 (0.01)	0.10	0.005 **
Functional health literacy				0.02 (0.01)	0.12	0.001 **	0.02 (0.01)	0.11	0.001 **
Self-efficacy				0.13 (0.03)	0.18	0.000 ***	0.12 (0.03)	0.16	0.000 ***
“I like to learn something new about health”				0.21 (0.03)	0.29	0.000 ***	0.18 (0.03)	0.24	0.000 ***
Perceived parental health orientation							0.13 (0.03)	0.18	0.000 ***
Corrected R^2^	0.020			0.164			0.193		
F for change in R^2^	4.618			41.215			26.243		

**Notes.** Dependent variable: HLS-Child-Q15-DE mean scores. Sample size for the regression models n = 706; Abbreviations: B = unstandardized regression coefficients; SE B: standard error; *β* = standardized regression coefficients; ** = *p* < 0.01; *** = *p* < 0.001.

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
