# Peer review of "Exploring Associated Factors of Subjective Health Literacy in School-Aged Children"

_ijerph, 2020, doi:10.3390/ijerph17051720_

Round 1

Reviewer 1 Report

Health literacy has been acknowledged as a determinant of health and a critical means of increasing health equity, so it is meaningful and interesting to investigate the relationships between different factors and (subjective) health literacy. This paper aims to research and explore potential determinants of subjective health literacy specifically for grade-four children. The presentation and organization of the materials, methods, and findings are clear and convictive. The discussion part of the research also provides details and explanations when I was reading the first part of the manuscript. It is joyful to read such a good article. Thanks to the authors.

I only have one minor suggestion or improvement: Currently, predictors are grouped or categorized into Demographics, Functional health literacy, Self-efficacy, etc. If consider to investigate dependence between any combinations of predictors and the health literacy, that may provide more interesting and expected results. E.g., maybe, family affluence and health knowledge as a pair is a strong determinant of health literacy, but not single one of them.

Thanks.

Author Response

Many thanks for the encouraging and uplifting feedback! The analysis you have pointed out is indeed interesting. In the beginning we explored different ways of categorizing the variables into the hierarchical models. We also explored a model where, family affluence and functional health literacy, together with age and gender were considered in the first model. The results were quite similar and these variables did not account for more explained variation in health literacy. Based on your comment, we have run quick analysis with family affluence and health knowledge as interaction term. Still not much difference can be seen. The standardized Beta for “affluence X knowledge” were as follows: .13 in the first Model and .10 in the third Model. The explained variance in the third model improved slightly to 22,5%.

Since in general there is so little known about the health literacy of children, our analysis is explorative and aims to answer basic questions firstly. Especially in case of affluence and knowledge it is already known that they might independently be related to health literacy. Therefore, it was important to us to answer this question at first, before turning to more complex interactive relationships.

 For future publications, we will keep your suggestion in mind and moreover use more suitable methods of answering those questions (e.g. path analysis).

Best wishes!

Reviewer 2 Report

The subject of the research is of interest considering that there is very little evidence on health literacy in children and adolescents.

The whole manuscript is presented clearly, the results are understandable and in accordance with the stated goals. The limitations of the study are appropriately expressed.

In my opinion, the most important limitations are two-fold. On the one hand, the design itself is problematic and even more so being as it is a survey of children and adolescents who can be very changeable depending on the moment. On the other hand, the questions themselves, for example in studying the construct of the perception of self-efficacy of a very young population with a single question, have many limitations. This is also seen with the study of motivation on health knowledge.

The authors may be able to explain more about this.

Author Response

Many thanks for the positive feedback! Indeed studying health literacy of such a young age group is tricky, since the stability of their answers is not well-known. For this first explorative study, we did not have the possibility to undertake a longitudinal design to verify how reliable the answers are over time. We acknowledge that it is difficult to study these constructs in such a young age group, however we believe that it is valuable to depict the opinions of children regarding different health-related matters (even if they might be biased). We also stated that we analyze the subjective perceptions (regarding health literacy) of children, therefore the results should be interpreted only in this realm. We hope this answer is satisfactory or otherwise look forward to specific recommendations on how to improve the manuscript!

Best wishes!

Reviewer 3 Report

Thank you for the opportunity to review this very interesting manuscript entitled "Exploring Associated Factors of Subjective Health Literacy in School-Aged Children".  The authors have done an effort to analyze the influence of the variables in the Children’s Subjective Health Literacy. However, I consider that this manuscript should be accepted after major revisions that I specify in these comments. The main change that I consider is the introduction section.

Title:

The title is concise, specific and relevant. Readers can understand the general meaning of the manuscript.

Abstract:

The abstract should be a total of about 200 words maximum. This manuscript has 237 words. The authors should reduce this abstract according the instructions for the authors of this journal. Also, these instructions for the authors specify “The abstract should be a single paragraph and should follow the style of structured abstracts, but without headings”. I suggest the authors re-write the abstract without the headings.

Introduction:

The introduction is very short in comparation of the other sections in this manuscript. The introduction should be more focused on the international studies about the subjective health literacy. I recommend including the definition of the health literacy in the beginning of the manuscript.

It is necessary a deeper introduction in some variables taking into account more studies to show the previous evidence in this area. In this section the authors should explain more previous studies about the factors that they analyzed in this study, and the influence of these variables on the subjective health literacy.

The specific objectives and hypothesis should be more concrete. The authors describe a general hypothesis, they must specify more the results that they hypothesized a priori. For example, the variables that they believe more predictive in the children’s health literacy.

Method:

Sample and procedure description: I suggest authors to rewrite this section. How many classrooms were selected in each school? How long did the data collection process take overall? How many time did the participants employ to fill the survey?

Also, the authors should specify better the sampling technique used in this study.

In the instruments section the authors should explain better the scale that they used for Functional health literacy. If they adapted the instrument for adults, the authors should specify the scale and their items.

Discussion:

I consider that the discussion in this section are correct. The authors discuss the results according previous studies in the area. However, I consider that the introduction is very short in comparation of this section.

I suggest including the possible future researches according to the results of this study. This it will be a positive aspect to continue the study in this area.

References:

According the instructions of this journal: references must be numbered in order of appearance in the text (including table captions and figure legends) and listed individually at the end of the manuscript. These references are correct. I recommend that the authors check these references after the changes in the introduction section.

Author Response

Many thanks for the constructive and precise feedback! We have reduced the abstract to under 200 words, expanded the introduction section by adding a definition of health literacy and making referrals to other relevant studies, analyzing the health literacy of young people.

Since data on children’s health literacy is extremely scare, we did not employ hypotheses regarding the magnitude of relationship between the independent variables and health literacy. Due to the lack of other data, we felt that we cannot make more concrete assumptions. Our hypothesis was only that the included factors might be related to health literacy. Why we picked those exact ones, we tried to explain in the methods section in the part regarding the multivariate analysis. Please let us know if you feel this needs further improvement.

We have provided a more detailed description of the study design, sampling technique, and of the measure regarding functional health literacy. We hope we incorporated the information adequately!

Best wishes

Round 2

Reviewer 3 Report

Thank you for the opportunity to review this very interesting manuscript entitled "Exploring Associated Factors of Subjective Health Literacy in School-Aged Children". The authors have done an effort to check and modify according to my suggests. Taking account this, like a reviewer of this manuscript now I consider that this manuscript could be published.